# Mooring Drag Effects in Interaction Problems of Waves and Moored Underwater Floating Structures

**Cheng-Tsung Chen [1,2], Jaw-Fang Lee [3,\*] and Chun-Han Lo [4]**

[1]  Department of Civil Engineering, Yancheng Institute of Technology, Yancheng 224051, China;
    ctchen42@yahoo.com.tw
[2]  Center for Innovative Research on Aging Society, National Chung Cheng University, Chia-yi 621, Taiwan
[3]  Department of Hydraulic and Ocean Engineering, National Cheng Kung University, Tainan 701, Taiwan
[4]  Water Resources Planning Institute, Water Resources Agency, Ministry of Economic Affairs,
    Taichung City 413, Taiwan; a6bennylo@gmail.com
\*  Correspondence: jflee@mail.ncku.edu.tw

**Abstract:** In contrast to either considering structures with full degrees of freedom but with wave force on mooring lines neglected or with wave scattering and radiation neglected, in this paper, a new analytic solution is presented for wave interaction with moored structures of full degrees of freedom and with wave forces acting on mooring lines considered. The linear potential wave theory is applied to solve the wave problem. The wave fields are expressed as superposition of scattering and radiation waves. Wave forces acting on the mooring lines are calculated using the Morison equation with relative motions. A coupling formulation among water waves, underwater floating structure, and mooring lines are presented. The principle of energy conservation, as well as numerical results, are used to verify the present solution. With complete considerations of interactions among waves and moored structures, the characteristics of motions of the structure, the wave fields, and the wave forces acting on the mooring lines are investigated.

**Keywords:** analytic solution; water waves; underwater floating structure; mooring forces; interaction

## 1. Introduction

With increasing development of ocean wave energy extraction, various types of underwater ocean structures were used in these aspects [1]. As incident waves acting on underwater floating structures, the wave forces in fact act on both main floating objects as well as the mooring deployments. The reactive motions of the structure systems then feedback into the surrounding wave fields, thus forming wave and structure interaction systems. The floating structures offer scattering and radiation phenomena on water waves, whereas wave forces on mooring lines induce motions of the mooring lines and cause motions of the floating structures and surrounding wave fields. In literature, most studies on interactions of floating structures and incident waves consider either problems neglecting wave forces on mooring lines [2] or problems with wave forces on mooring lines but without scattering and radiation structural effects [3].

Numerical simulations are commonly used to calculate problems of ocean structures subjected to incident waves. A three-dimensional finite element method was developed by Huang et al. [4] to calculate wave diffraction, wave radiation, and body responses of multiple bodies of arbitrary shape. Sannasiraj et al. [5] applied both experimental and finite element methods to study behaviors of pontoon-type floating structures in waves. The slack mooring lines were idealized as spring coefficients calculated from the catenary equation of cables. Chen et al. [6] used a boundary integral and Green's function to investigate floating breakwaters consisting of rectangular pontoon and horizontal plates.

The mooring lines were calculated using the static catenary equation. Mohapatra and Sahoo [7] used a Green's integral to study the interaction problem of oblique surface gravity waves with a floating flexible plate. In Cao and Zhao [8], a Computational fluid dynamics (CFD) numerical method was used to study nonlinear dynamic behaviors of a two-dimensional box-shaped floating structure in focused waves. Mohapatra and Soares [9] used linearized Boussinesq equations to study the wave forces acting on a floating structure over a flat bottom. Kao et al. [10] used a boundary element method to solve the problem of floating structures subjected to incident waves. Rivera-Arreba [11] studied the dynamic response of the floating wind turbine subjected to wind and waves. On the other hand, Guo et al. [12] used a linear wave theory to calculate the wave forces acting on the structure. In numerical simulations for waves acting on floating structures with moorings, mostly the mooring mechanisms were included and wave interferences were considered. However, wave forces acting on mooring lines were not considered. The reason was that the calculation of wave forces on mooring lines includes wave kinematics and motions of mooring lines, which complicated mathematical formulation in the problem.

As for the analytic approach solving for interaction problems of wave and mooring floating structures, in the analysis of waves interacting with moored floating structures, most researches neglected wave forces acting on the mooring lines [2]. Lee [13] first considered a tension-leg floating structure subjected to wave actions, in which the floating structure was assumed to have surge motion only. The wave force acting on the tension leg was calculated by a linearized Morison equation, and an analytic solution was proposed for the entire interacted problem. Lee et al. [14] applied the interaction methodology of large and small structures, extending to tethered mooring tension leg floating structures. Lee and Wang [15] extended the same technique to problems of tension leg platforms with net cages. In the articles mentioned above, the floating structures allow only surge motion; therefore, analytic solutions could be obtained without any difficulty.

If the structures are allowed to include heave and surge motions, one would then encounter the typical heave radiation problem. Lee [16] proposed an easy-to-follow derivation to obtain an analytic solution. Other than that, a particular solution approach has to be applied that was not convenient to use in obtaining the solution. It could be said that, using Lee's method, the radiation problem of the two-dimensional structure can be obtained completely. Chen et al. [17] then investigated the problem of wave interaction with a floating structure with moorings, and wave forces acting on the mooring lines were included. In the two-dimensional problem, the floating structure had complete three degrees of freedom, namely, surge, heave, and pitch.

In this study, an underwater floating structure with moorings subjected to incident waves is considered. Zheng et al. [18] presented an analytic solution for oblique waves passing an underwater floating rectangular structure. A particular solution was used to satisfy the nonhomogeneous boundary value problem. However, the analytic solution could not be simplified to the case of normal incident wave, as in the two-dimensional problem. The intention of this paper is to present a new analytic solution to the problem. The significance of this paper is that the wave forces acting on the mooring lines are considered in the coupling problems of waves and floating structures and the problems solved analytically. The floating structure has motions with three degrees of freedom. The effects of the mooring lines subjected to wave forces on the hydrodynamics of the wave and structure interaction system are investigated.

## 2. Problem Description and Solution

The problem considered is an underwater floating structure moored to the sea bottom and subjected to the action of incident waves, as shown in Figure 1. A Cartesian coordinate system is adopted with the positive x pointed to the right and positive z pointed upward. The constant water depth is $h$, the width of the structure is $2\ell$, the structural submergence is $d_1$, and the distance from the structural bottom to the sea bottom is $d_2$. The incident wave $\eta^I$ is propagating from $-x$ to the $+x$ direction. With the action of incident waves, the structure system does respond accordingly and also

interferes with the surrounding wave field. In this two-dimensional problem, the floating structure has, in general, three degrees of freedom, namely, surge, heave, and pitch motions. With the prerequisite periodic motion, the displacement functions of the structure can be expressed as

$$\xi_j = s_j \cdot e^{-i\omega t}, \ j = 1, 2, 3 \tag{1}$$

where the subscripts 1, 2, and 3 represent surge, heave, and pitch, respectively. $s_j$ represents amplitudes of the structural motions. Since an analytic solution is pursued, and with a rectangular shape of the structure, the method of separation of variables is used to solve the problem, and the domain is divided into four regions, as indicated in Figure 1. Region 1 is in front of the structure, regions 2 and 4 are above and beneath the structure, and region 3 is behind the structure.

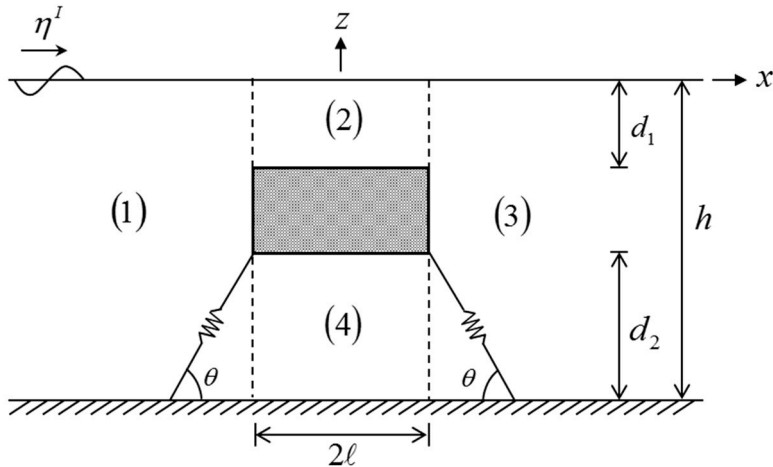

**Figure 1.** Definition sketch of waves incidents on an underwater floating structure with moorings.

The interference wave field surrounding the floating structure needs to be solved, in addition to the known incident wave, so that wave forces acting on the floating structure and the moorings can be calculated, and so, to calculate structural motions.

A linear potential wave theory is used to describe the wave problem. The definition of the velocity $\vec{V}$ related to the wave potential function $\Phi$ is written as

$$\vec{V} = -\nabla\Phi \tag{2}$$

where $\nabla$ is the gradient operator. Since steady and periodic problems are considered, the periodic time function can be factored out and wave potential be expressed as

$$\Phi(x, z, t) = \phi(x, z) \cdot e^{-i\omega t} \tag{3}$$

where $\omega = 2\pi/T$, $T$ is the wave period, and $i = \sqrt{-1}$. The incident wave potential is given as

$$\Phi^I(x, z, t) = \frac{igA^I}{\omega} \cdot \frac{\cosh K(z + h)}{\cosh Kh} \cdot e^{i (Kx - \omega t)} \tag{4}$$

where $g$ is the gravitational constant, $A^I$ is the wave amplitude, and $K$ is the wave number that is calculated by the dispersion equation

$$\omega^2 = gK \tanh Kh \tag{5}$$

The entire problem is decomposed into a scattering problem and radiation problems in the three degrees of freedom [2]; i.e., the interference wave potential is expressed as

$$\Phi(x,z,t) = \Phi^D + \sum_{j=1}^{3} s_j \cdot \Phi^j \tag{6}$$

where $\Phi^D$ is the scattering wave produced by the incident wave acting on the structure with the structure held fixed. On the other hand, $\Phi^j$ corresponds to the radiation wave for the structure having motion in the $j$-direction and with unit amplitude. For the problem here, surge, heave, and pitch motions.

Now, the task becomes to obtain analytical solutions for the scattering wave and the radiation waves and, particularly, the solution expression for each individual region. Since analytic solutions for the problem of a surface-floating structure have been presented by Chen et al. [17], the solutions for divided regions can be followed, except the region 2 for heave and pitch radiation problems. Further, solutions for the pitch radiation problem are an application of heave and surge problems; therefore, only derivation details of the region 2 in the heave problem will be shown here.

The boundary value problem for region 2 in the heave radiation problem can be written as:

The governing equation:

$$\nabla^2 \phi_2^2 = 0, \quad -d_1 < z < 0, \quad -\ell < x < \ell \tag{7}$$

The upper free surface condition:

$$\frac{\partial \phi_2^2}{\partial z} = \frac{\omega^2}{g} \phi_2^2, \quad z = 0 \tag{8}$$

The lower boundary condition:

$$\frac{\partial \phi_2^2}{\partial z} = i\omega s_2, \quad z = -d_1 \tag{9}$$

The left boundary condition:

$$\phi_2^2 = \phi_1^2, \quad x = -\ell \tag{10}$$

The right boundary condition:

$$\phi_2^2 = \phi_3^2, \quad x = \ell \tag{11}$$

Note that it is the nonhomogeneous form shown in Equation (1) that poses the difficulty in obtaining the solution. In this study, a method proposed by Lee [6] is used to derive the solution. In Equations (7)–(11), the wave potential is divided into two parts:

$$\phi_2^2 = \widetilde{\phi}_2^2 + \hat{\phi}_2^2 \tag{12}$$

where $\widetilde{\phi}_2^2$ and $\hat{\phi}_2^2$ satisfy vertically and horizontally homogeneous conditions, respectively, as shown in Figure 2.

Following the standard method of separation of variables, one can obtain the solutions

$$\begin{aligned}
\phi_2^2 &= \widetilde{\phi}_2^2 + \hat{\phi}_2^2 \\
&= \sum_{n=0}^{\infty} \left[ A_{2n}^2 e^{-k_{2n}(x+\ell)} + B_{2n}^2 e^{k_{2n}(x-\ell)} \right] \cos[k_{2n}(z+d_1)] \\
&\quad + \sum_{n=1}^{\infty} D_{2n}^2 \left( \mu_{n1} e^{\gamma_n(z-h)} + \mu_{n2} e^{-\gamma_n(z+h)} \right) \sin \gamma_n(x+\ell)
\end{aligned} \tag{13}$$

in which the coefficients $k_{2n}$, $\gamma_n$, $\mu_{n1}$, $\mu_{n2}$, and $D^2_{2n}$ are given in Appendix A. The way of obtaining the solution for region 2 in the heave radiation problem can also be applied to obtain the solution for the same region 2 in the pitch radiation problem.

$$\frac{\partial \phi_2^2}{\partial z} = \frac{\omega^2}{g}\phi_2^2$$

$$\phi_1^2 = \phi_2^2 \quad \boxed{(2)} \quad \phi_2^2 = \phi_3^2$$

$$\frac{\partial \phi_2^2}{\partial z} = i\omega s_2$$

$$= \quad \phi_1^2 = \widetilde{\phi}_2^2 \quad \boxed{\begin{array}{c}\dfrac{\partial \widetilde{\phi}_2^2}{\partial z} = \dfrac{\omega^2}{g}\widetilde{\phi}_2^2 \\ (2) \\ \dfrac{\partial \widetilde{\phi}_2^2}{\partial z} = 0\end{array}} \quad \widetilde{\phi}_2^2 = \phi_3^2 \quad + \quad \hat{\phi}_2^2 = 0 \quad \boxed{\begin{array}{c}\dfrac{\partial \hat{\phi}_2^2}{\partial z} = \dfrac{\omega^2}{g}\hat{\phi}_2^2 \\ (2) \\ \dfrac{\partial \hat{\phi}_2^2}{\partial z} = i\omega s_2\end{array}} \quad \hat{\phi}_2^2 = 0$$

**Figure 2.** Decomposition of nonhomogeneous boundary conditions.

As for the rest of the scattering and the radiation problems, one can easily obtain the solutions. For completeness, they are also listed here. For the wave scattering problem, solutions for the four regions can be written as:

$$\phi_1^D = \sum_{n=0}^{\infty} B_{1n}^D \cos[k_n(z+h)]e^{k_n(x+\ell)} \tag{14}$$

$$\phi_2^D = \sum_{n=0}^{\infty} \left[A_{2n}^D e^{-k_{2n}(x+\ell)} + B_{2n}^D e^{k_{2n}(x-\ell)}\right] \cos[k_{2n}(z+d_1)] \tag{15}$$

$$\phi_3^D = \sum_{n=0}^{\infty} A_{3n}^D \cos[k_n(z+h)]e^{-k_n(x-\ell)} \tag{16}$$

$$\phi_4^D = \left(A_{40}^D x + B_{40}^D\right) + \sum_{n=1}^{\infty} \left[A_{4n}^D e^{-k_{4n}(x+\ell)} + B_{4n}^D e^{k_{4n}(x-\ell)}\right] \cos[k_{4n}(z+h)] \tag{17}$$

For the radiation wave problems, the three radiations surge, heave, and pitch are expressed, respectively, as:

Surge radiation:

$$\phi_1^1 = \sum_{n=0}^{\infty} B_{1n}^1 \cos[k_n(z+h)]e^{k_n(x+\ell)} \tag{18}$$

$$\phi_2^1 = \sum_{n=0}^{\infty} \left[A_{2n}^1 e^{-k_{2n}(x+\ell)} + B_{2n}^1 e^{k_{2n}(x-\ell)}\right] \cos[k_{2n}(z+d_1)] \tag{19}$$

$$\phi_3^1 = \sum_{n=0}^{\infty} A_{3n}^1 \cos[k_n(z+h)]e^{-k_n(x-\ell)} \tag{20}$$

$$\phi_4^1 = \left(A_{40}^1 x + B_{40}^1\right) + \sum_{n=1}^{\infty} \left[A_{4n}^1 e^{-k_{4n}(x+\ell)} + B_{4n}^1 e^{k_{4n}(x-\ell)}\right] \cos[k_{4n}(z+h)] \tag{21}$$

Heave radiation:

$$\phi_1^2 = \sum_{n=0}^{\infty} B_{1n}^2 \cos[k_n(z+h)]e^{k_n(x+\ell)} \tag{22}$$

$$\phi_2^2 = \sum_{n=0}^{\infty} \left[A_{2n}^2 e^{-k_{2n}(x+\ell)} + B_{2n}^2 e^{k_{2n}(x-\ell)}\right] \cos[k_{2n}(z+d_1)]$$
$$+ \sum_{n=1}^{\infty} D_{2n}^2 \left(\mu_{n1} e^{\gamma_n(z-h)} + \mu_{n2} e^{-\gamma_n(z+h)}\right) \sin \gamma_n(x+\ell) \tag{23}$$

$$\phi_3^2 = \sum_{n=0}^{\infty} A_{3n}^2 \cos[k_n(z+h)]e^{-k_n(x-\ell)} \tag{24}$$

$$\phi_4^2 = \left(A_{40}^2 x + B_{40}^2\right) + \sum_{n=1}^{\infty} \left[A_{4n}^2 e^{-k_{4n}(x+\ell)} + B_{4n}^2 e^{k_{4n}(x-\ell)}\right] \cos[k_{4n}(z+h)]$$
$$+ \sum_{n=1}^{\infty} F_{4n}^2 \cosh \gamma_n(z+h) \sin \gamma_n(x+\ell) \tag{25}$$

Pitch radiation:

$$\phi_1^3 = \sum_{n=0}^{\infty} B_{1n}^3 \cos[k_n(z+h)]e^{k_n(x+\ell)} \tag{26}$$

$$\phi_2^3 = \sum_{n=0}^{\infty} \left[A_{2n}^3 e^{-k_{2n}(x+\ell)} + B_{2n}^2 e^{k_{2n}(x-\ell)}\right] \cos[k_{2n}(z+d_1)]$$
$$+ \sum_{n=1}^{\infty} D_{2n}^3 \left(\mu_{n1} e^{\gamma_n(z-h)} + \mu_{n2} e^{-\gamma_n(z+h)}\right) \sin \gamma_n(x+\ell) \tag{27}$$

$$\phi_3^3 = \sum_{n=0}^{\infty} A_{3n}^3 \cos[k_n(z+h)]e^{-k_n(x-\ell)} \tag{28}$$

$$\phi_4^3 = \left(A_{40}^3 x + B_{40}^3\right) + \sum_{n=1}^{\infty} \left[A_{4n}^3 e^{-k_{4n}(x+\ell)} + B_{4n}^3 e^{k_{4n}(x-\ell)}\right] \cos[k_{4n}(z+h)]$$
$$+ \sum_{n=1}^{\infty} F_{4n}^3 \cosh \gamma_n(z+h) \sin \gamma_n(x+\ell) \tag{29}$$

where $k_n$, $k_{4n}$, $F_{4n}^2$, $D_{2n}^3$, and $F_{4n}^3$ are listed in Appendix A. The undetermined coefficient shown in Equations (18)–(29) are then obtained by solving simultaneous equations obtained from matching the velocity and pressure conditions at the interfacial boundary of two neighboring regions and integration of associated water depth multiplied by the orthogonal functions.

Once the decomposed wave scattering problem and the wave radiation problem of unit amplitude are obtained, the unknown variables shown in the interference wave potential, Equation (6), are amplitudes of the structural motion, which can then be solved by the equations of motion of the structure.

The equations of motion of the underwater floating structure can be written as [19]:

$$[M] \left\{ \begin{array}{c} \ddot{\xi}_1 \\ \ddot{\xi}_2 \\ \ddot{\xi}_3 \end{array} \right\} = \left\{ \begin{array}{c} F_1 \\ F_2 \\ F_3 \end{array} \right\} - \left\{ \begin{array}{c} T_1 \\ T_2 \\ T_3 \end{array} \right\} + \left\{ \begin{array}{c} F_1^M \\ F_2^M \\ F_3^M \end{array} \right\} \tag{30}$$

where $[M]$ is the mass matrix, $\{F\}$ is the wave forces acting on the floating structure, $\{T\}$ represents the restoring force of the mooring springs, and $\{F^M\}$ is the wave forces acting on the mooring lines. The mass matrix can be expressed as:

$$[M] = \begin{bmatrix} m & 0 & 0 \\ 0 & m & 0 \\ 0 & 0 & I_0 \end{bmatrix} \tag{31}$$

in which $m$ is mass of the structure and $I_0$ is moment of inertia.

Wave forces acting on the floating structure can be calculated using wave potentials surrounding the structure, and be expressed as:

$$\left\{ \begin{array}{c} F_1 \\ F_2 \\ F_3 \end{array} \right\} = \left[ f^R \right] \left\{ \begin{array}{c} s_1 \\ s_2 \\ s_3 \end{array} \right\} + \left\{ \begin{array}{c} f_1^D \\ f_3^D \\ f_3^D \end{array} \right\} \tag{32}$$

where $\left[ f^R \right]$ and $\left\{ f^D \right\}$ are calculated from radiated potentials of unit amplitudes and diffracted potentials, respectively. Detailed expressions are given in Appendix B.

The restoring forces produced by the mooring springs can be calculated according to orientations of the springs $\overline{AB}$ and $\overline{CD}$, and be expressed as:

$$T^{\overline{AB}} = \left[ \begin{array}{ccc} K_{11}^{\overline{AB}} & K_{12}^{\overline{AB}} & K_{13}^{\overline{AB}} \\ K_{21}^{\overline{AB}} & K_{22}^{\overline{AB}} & K_{23}^{\overline{AB}} \\ K_{31}^{\overline{AB}} & K_{32}^{\overline{AB}} & K_{33}^{\overline{AB}} \end{array} \right] \left[ \begin{array}{c} \xi_1 \\ \xi_2 \\ \xi_3 \end{array} \right] \tag{33}$$

$$T^{\overline{CD}} = \left[ \begin{array}{ccc} K_{11}^{\overline{CD}} & K_{12}^{\overline{CD}} & K_{13}^{\overline{CD}} \\ K_{21}^{\overline{CD}} & K_{22}^{\overline{CD}} & K_{23}^{\overline{CD}} \\ K_{31}^{\overline{CD}} & K_{32}^{\overline{CD}} & K_{33}^{\overline{CD}} \end{array} \right] \left[ \begin{array}{c} \xi_1 \\ \xi_2 \\ \xi_3 \end{array} \right] \tag{34}$$

in which the expressions of the stiffness matrices are given in Appendix C.

The wave forces acting on the mooring lines are calculated using a linearized Morison [3]. Using the present analytic solutions for the wave fields and the associated geometrical deployments of the mooring lines, the wave forces can be calculated. Detailed derivations are given in Appendix D. The induced forces acting on the floating structure can then be written as

$$\left\{ \begin{array}{c} F_1^M \\ F_2^M \\ F_3^M \end{array} \right\} = \left[ f^{MR} \right] \left\{ \begin{array}{c} s_1 \\ s_2 \\ s_3 \end{array} \right\} + \left\{ \begin{array}{c} f_1^{MD} \\ f_3^{MD} \\ f_3^{MD} \end{array} \right\} \tag{35}$$

in which $\left[ f^{MR} \right]$ is the radiation wave generated coefficient matrix.

Having the required expressions of all forces acting on the floating structure, including scattering and radiation waves, mooring restoring forces, and effects of wave forces on mooring lines, the equations of motion of the structure, Equation (30), can be solved and expressed as:

$$\left\{ \begin{array}{c} s_1 \\ s_2 \\ s_3 \end{array} \right\} = \left[ \widetilde{K} \right]^{-1} \left( -i\omega\rho \left\{ \begin{array}{c} f_1^D \\ f_3^D \\ f_3^D \end{array} \right\} + \left\{ \begin{array}{c} f_1^{MD} \\ f_2^{MD} \\ f_3^{MD} \end{array} \right\} \right) \tag{36}$$

where the general stiffness matrix is

$$[\widetilde{K}] = \left( -\omega^2 [M] + i\omega\rho \left[ f^R \right] + [K] - \left[ f^{MR} \right] \right) \tag{37}$$

Once amplitudes of the structural motion can be calculated, then the wave potentials of the entire problem domain can then be determined via Equation (6). The reflected wave in front of the structure $\eta^R$ and the transmitted wave behind the structure $\eta^T$ can then be calculated using the Bernoulli's equation. So far, the entire coupling problem is solved. A consideration of wave forces calculated from incident wave, scattering wave, radiation wave, and wave forces on the mooring lines, then the motions of the structure with moorings, are solved.

## 3. Results and Discussion

In this paper, the problem of moored underwater floating structures with motions of full degrees of freedom subjected to incident waves is investigated, and an analytic solution is presented. The present analytic solution is first validated by conservation of wave energy with no energy dissipation. In the present theory, the only energy loss in the problem is the drag forces acting on the mooring lines; therefore, if the drag coefficient is specified zero, $C_D = 0$, then the energy conservation of the system should satisfy. Figure 3 shows reflection and transmission coefficients and total wave energy, $K_r^2 + K_t^2$, versus relative water depth, $Kh$, and as is expected, the wave energy conserved to unity. The conditions used are water depth, $h = 10$ m, and incident wave amplitude, $A^I = 0.5$ m; other parameters used are $d_1/h = 0.2$, $\ell/h = 0.5$, $a/h = 0.3$, and the virtual coefficient $C_M = 2.0$. The corresponding result for the case considering the drag coefficient, $C_D = 2.0$, is shown in Figure 4. It is reasonable to identify that, with the drag effect, the total energy indicates dissipation. With energy dissipation, the total energy curve decreases about 10% at resonant frequency, the reflection coefficient decreases from 1.0 to 0.94, and the transmission coefficient increases from zero to 0.13. The present analytic solution is further applied to calculate a wave scattering problem of an underwater plate, and the results compared with that calculated using a numerical finite element method (Cheong et al. [20]). The conditions used are water depth, $h = 10$ m, and incident wave amplitude, $A^I = 0.5$ m; other parameters used are $d_1/h = 0.3$, $\ell/h = 0.5$, and $a/h = 0.025$. The comparisons of the reflection and transmission coefficients versus dimensionless water depth are shown in Figure 5, in which good agreements are indicated.

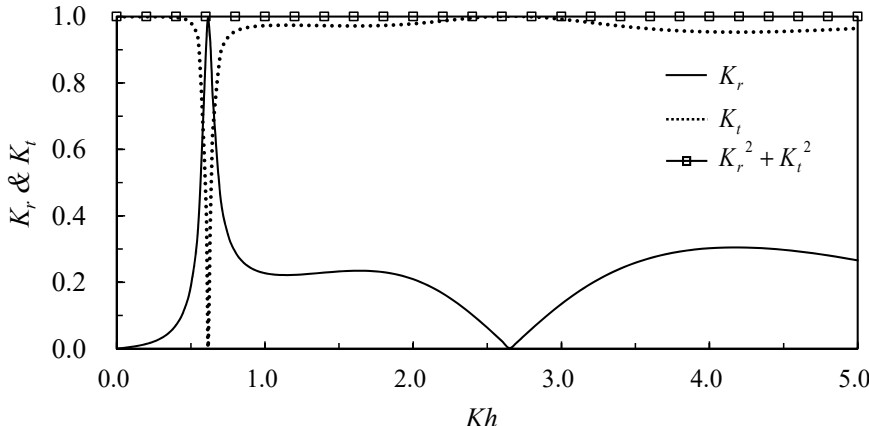

**Figure 3.** Reflection coefficient; transmission coefficient; and total energy versus relative water depth, $Kh$ ($C_D = 0.0$, $d_1/h = 0.2$, $\ell/h = 0.5$, $a/h = 0.3$, and $C_M = 2.0$).

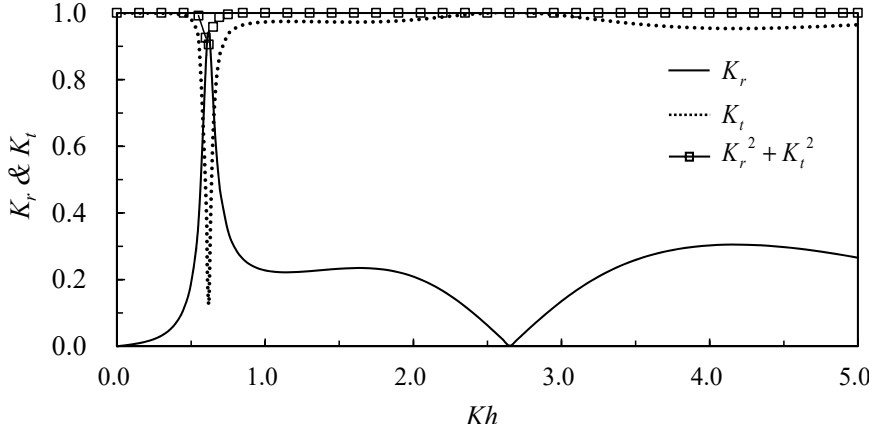

**Figure 4.** Reflection coefficient; transmission coefficient; and total energy versus relative water depth, $Kh$ ($C_D = 2.0$, $d_1/h = 0.2$, $\ell/h = 0.5$, $a/h = 0.3$, and $C_M = 2.0$).

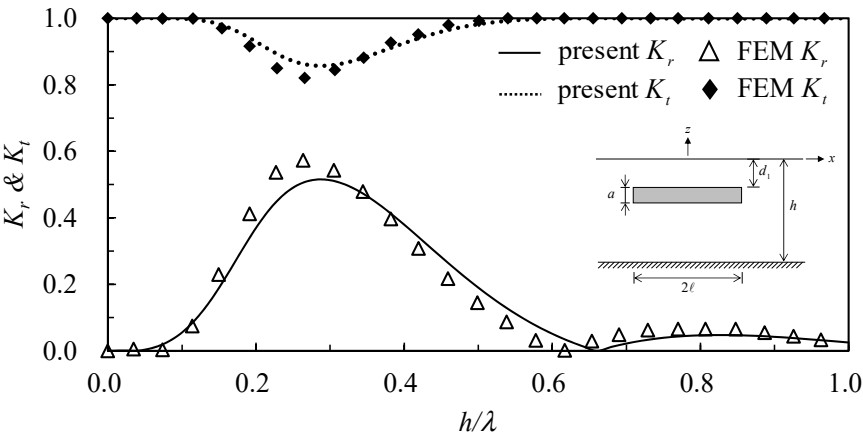

**Figure 5.** Reflection coefficient versus dimensionless water depth, $h/\lambda$, for a fixed underwater plate ($h = 10$ m, $d_1/h = 0.3$, $\ell/h = 0.5$, and $a/h = 0.025$).

In the present theory, the wave forces acting on the mooring lines are considered. To comply with motions of the mooring lines, in the wave force calculation, a relative flow velocity is used in the Morison equation. Since the motions of the mooring lines are not known a priori, an iteration algorithm is used in the calculation. Figure 6 shows the iteration times versus relative water depth, $Kh$. The conditions used are water depth, $h = 10$ m, and incident wave amplitude, $A^I = 0.5$ m; other parameters used are $d_1/h = 0.25$, $\ell/h = 0.2$, and $a/h = 0.3$, and the drag coefficient and the virtual mass coefficient, $C_D = 2.0$ and $C_M = 2.0$, respectively. The iteration number can be as high as 14 times at the resonant frequency and only one time at other frequencies. Additionally, with a higher drag coefficient, a higher iteration number is required.

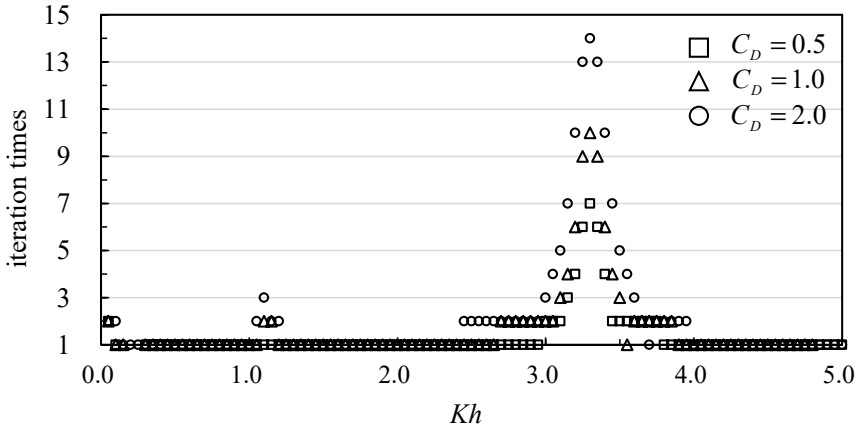

**Figure 6.** Iteration number versus relative water depth for different drag coefficients ($d_1/h = 0.25$, $\ell/h = 0.2$, $a/h = 0.3$, $C_D = 2.0$, and $C_M = 2.0$).

Using the same conditions as those in Figure 6, effects of the drag coefficient on wave reflection and wave transmission are shown in Figures 7 and 8, respectively. With the increase of the drag coefficient from zero to 2.0, the reflection coefficient at the first resonant peak at a lower frequency drops from 1.0 to 0.94, while the second peak at a higher frequency drops from 1.0 to 0.48. It is reasonable that the drag force acting on mooring lines can dampen only high-frequency short waves; rather, it is not sufficient in reducing wave energy of long waves. Therefore, high-frequency waves at resonant peaks are damped and decrease the reflection coefficient. The results also indicate that the increase of the drag coefficient from zero to 2.0 can dampen out 50% of the reflected waves. The tendency reverses for the transmission coefficient. The effects of the drag coefficient on structural motions are shown in Figures 9–11 for surge, heave, and pitch motions, respectively. Similar trends can be observed.

The increase of the drag coefficient reduces amplitudes of surge and pitch motions at high-frequency peaks, whereas it is not obvious at low-frequency resonant peaks. In this study, since it is difficult to obtain experimental results for comparison, numerical results using a boundary element method [21] for the case of $C_D = 2.0$ are also plotted for comparison. The comparisons also validate the present analytic solution for the problem.

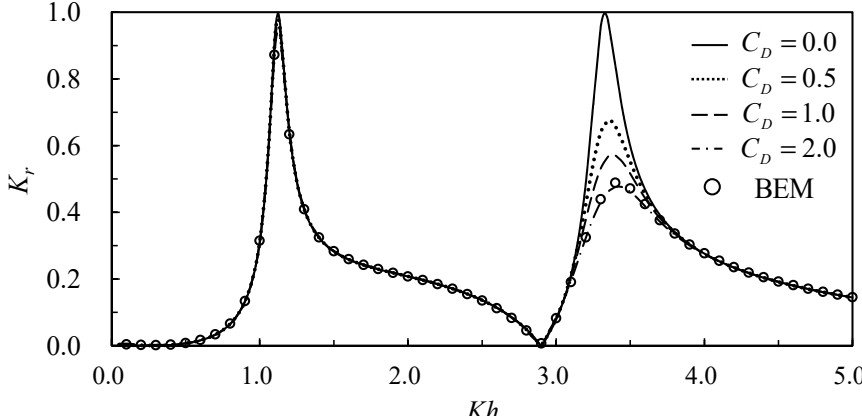

**Figure 7.** Reflection coefficient versus relative water depth for various drag coefficients ($d_1/h = 0.25$, $\ell/h = 0.2$, $a/h = 0.3$, and $C_M = 2.0$).

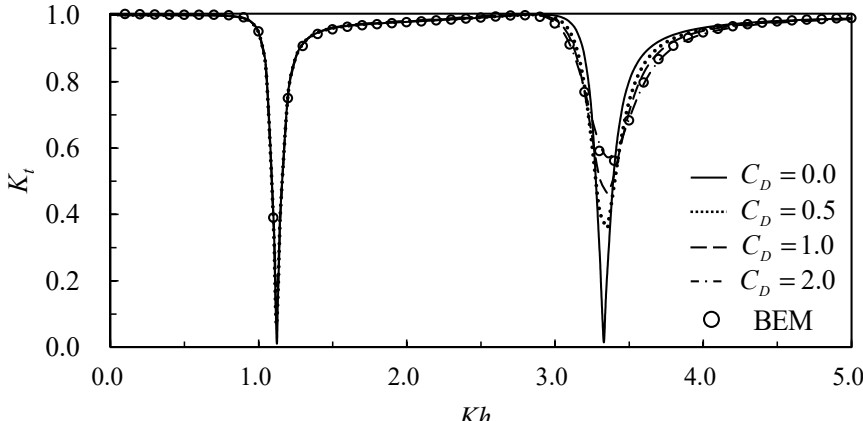

**Figure 8.** Transmission coefficient versus relative water depth for various drag coefficients ($d_1/h = 0.25$, $\ell/h = 0.2$, $a/h = 0.3$, and $C_M = 2.0$).

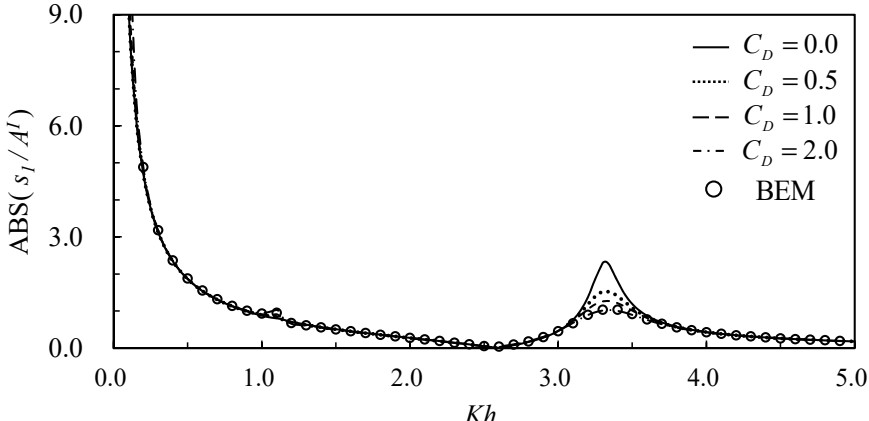

**Figure 9.** Surge motion amplitude versus relative water depth for various drag coefficients ($d_1/h = 0.25$, $\ell/h = 0.2$, $a/h = 0.3$, and $C_M = 2.0$).

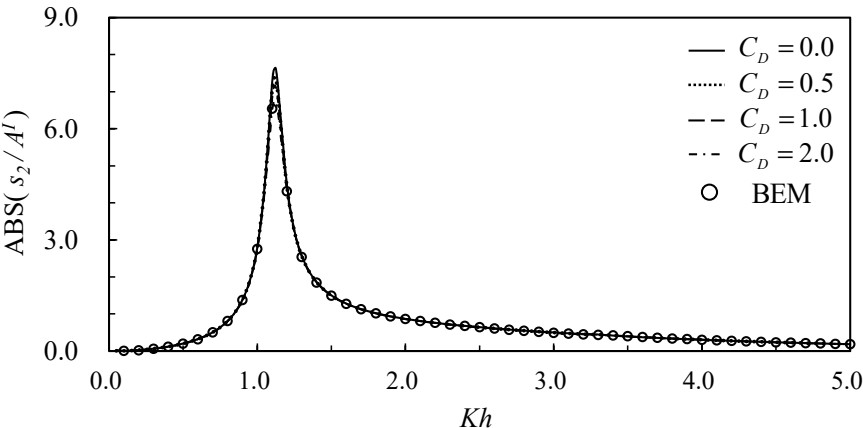

**Figure 10.** Heave motion amplitude versus relative water depth for various drag coefficients ($d_1/h = 0.25$, $\ell/h = 0.2$, $a/h = 0.3$, and $C_M = 2.0$).

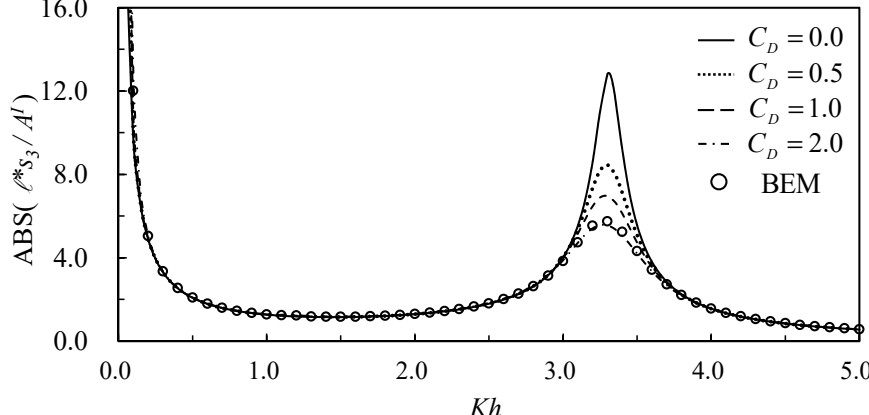

**Figure 11.** Pitch motion amplitude versus relative water depth for various drag coefficients ($d_1/h = 0.25$, $\ell/h = 0.2$, $a/h = 0.3$, and $C_M = 2.0$).

Figure 12 shows dimensionless horizontal and vertical forces versus relative water depth, that the horizontal force is divided by $\rho g A^I a$ and the vertical force is divided by $2\rho g A^I \ell$. $F^M_{\overline{AB},1}$ represents the horizontal force, while $F^M_{\overline{AB},2}$ represents the vertical force on the spring $\overline{AB}$. $F^M_{\overline{CD},1}$ represents the horizontal force, while $F^M_{\overline{CD},2}$ represents the vertical force on the spring $\overline{CD}$. For the given conditions $d_1/h = 0.25$, $\ell/h = 0.2$, $a/h = 0.3$, $C_D = 2.0$, and $C_M = 2.0$, wave forces acting on the upwind mooring lines, $\overline{AB}$, are obviously smaller than the mooring line, $\overline{CD}$, on the lee side. Additionally, the horizontal component of the forces are bigger than vertical ones. The maximum wave forces can reach up to 12% of the incident wave forces.

The present analytic solution is used to study the submerged depth of the structure on reflection and transmission coefficients and motions of the structure. The conditions used are water depth, $h = 10$ m, incident wave amplitude, $A^I = 0.5$ m, and width and height of the structure, $\ell/h = 0.4$ and $a/h = 0.3$. The submerged depths considered are near the water surface; one-fourth the water depth; and one-half the water depth ($d_1/h = 0.10$, $d_1/h = 0.25$, and $d_1/h = 0.50$). The dimensionless water depth related to the wave length covers the range from shallow water, $Kh < \pi/10$, up to the deep water, $Kh < \pi$. Variations of the reflection coefficient and the transmission coefficient versus $Kh$ for various structural submergences $d_1/h = 0.10$, $0.25$, and $0.50$ are shown in Figures 13 and 14, respectively. It can be expected that, since the underwater structure is blocking the incident wave while located under the water surface, the nearer the structure is close to the water surface, the structure can block more surface waves, except the longer waves can have less effect from the structure. Figure 13 also

shows the structure can have a total reflection for short waves and near the free surface. Furthermore, at the resonant frequency, there exists a total reflection. Figure 14 indicates a reverse tendency for the transmission coefficient versus dimensionless water depth.

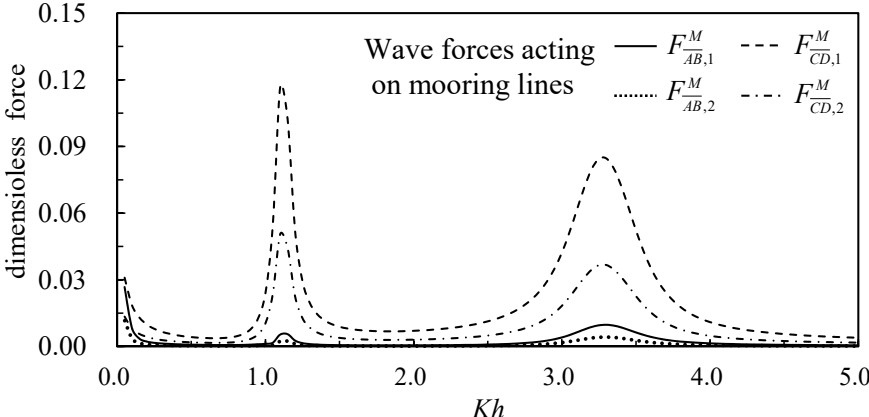

**Figure 12.** Wave forces on the mooring lines versus relative water depth ($d_1/h = 0.25$, $\ell/h = 0.2$, $a/h = 0.3$, $C_D = 2.0$, and $C_M = 2.0$).

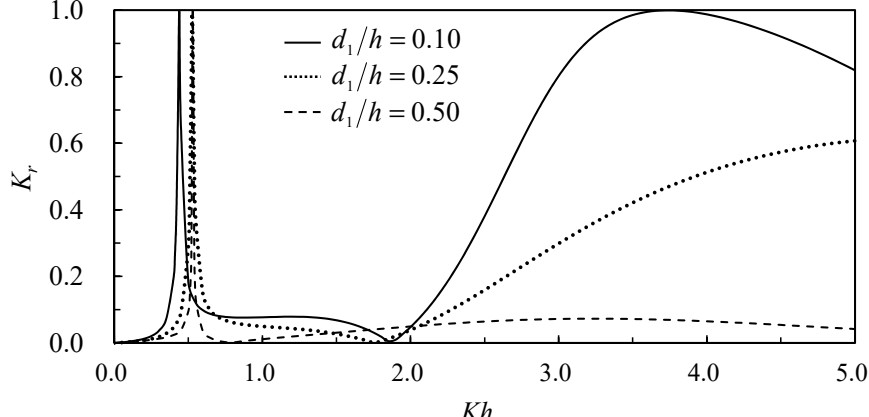

**Figure 13.** Reflection coefficient versus relative water depth for various submerged depths of the structure ($d_1/h = 0.10$, 0.25, and 0.50).

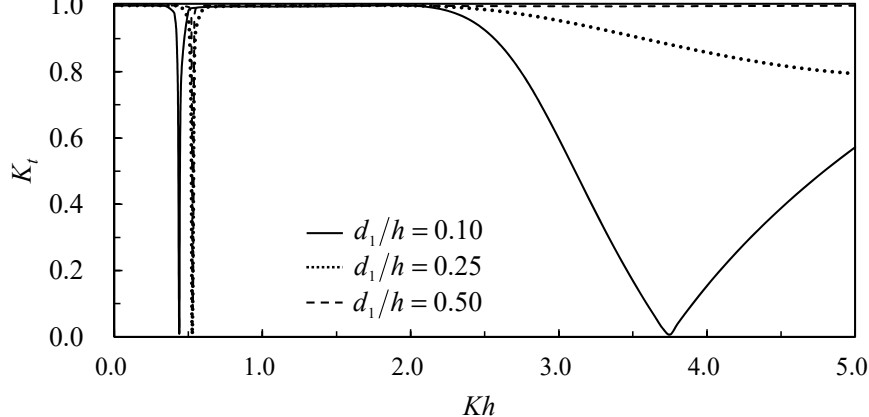

**Figure 14.** Transmission coefficient versus relative water depth for various submerged depths of the structure ($d_1/h = 0.10$, 0.25, and 0.50).

Effects of various submerged depths of the structure on the motions of the structure, namely, surge, heave, and pitch, are shown in Figures 15–17, respectively. The surge and pitch amplitudes

decrease with the increasing relative water depth (the shorter waves), as the shorter waves induce less structural motions. In general, the structure located near the free surface can get bigger motions. The same tendency applies to the heave motion, but there exists a resonant frequency due to the hydrostatic restoring force of the water buoyancy. The resonant frequency shifted for different structural submergence due to different hydrodynamic forces acting on the structure.

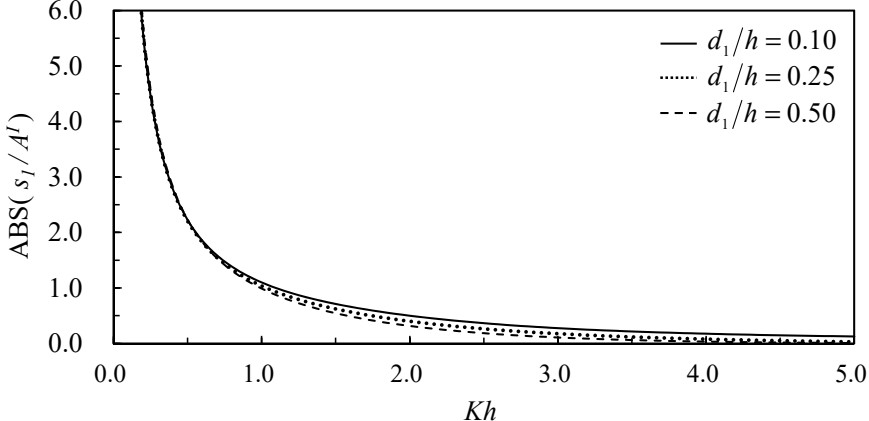

**Figure 15.** Dimensionless surge amplitude versus relative water depth for various submerged depths of the structure ($d_1/h = 0.10$, 0.25, and 0.50).

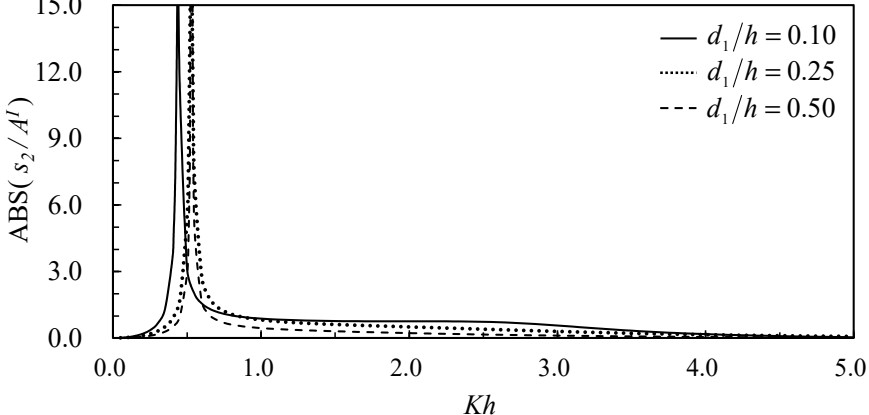

**Figure 16.** Heave amplitude versus relative water depth for various submerged depths of the structure ($d_1/h = 0.10$, 0.25, and 0.50).

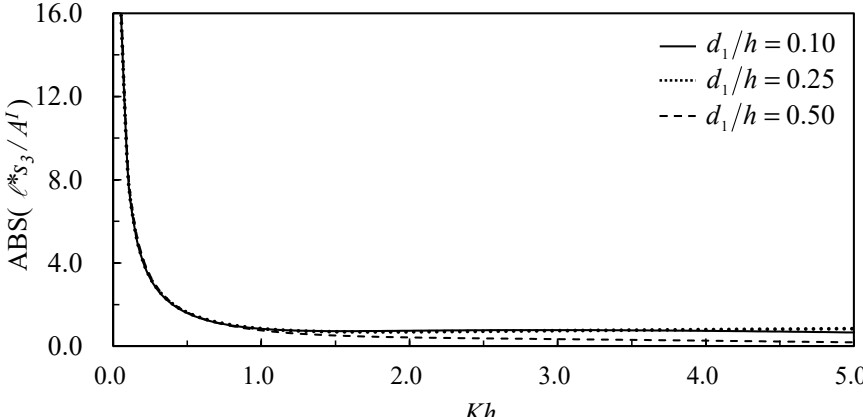

**Figure 17.** Pitch amplitude versus relative water depth for various submerged depths of the structure ($d_1/h = 0.10$, 0.25, and 0.50).

In this study, we emphasize that the moored underwater structures with motions of full degrees of freedom subjected to actions of incident waves and present an analytic solution. However, the solution is restricted to geometrical deployment and a linear assumption. Nevertheless, the methodology can be expanded to nonlinear problems via a higher-order solution; the linear solution can be applied as a preliminary identification of the characteristics of the practical problems.

## 4. Conclusions

With considerations of wave forces acting on mooring lines, a new analytic solution is presented for the problem of an underwater moored floating structure with motions of full degrees of freedom subjected to incident waves. A coupling formulation among water waves, underwater floating structure, and mooring lines is presented. Iterations for the drag coefficients, energy conservation without drag loss, reflection and transmission coefficients, and comparisons of wave scatterings with a finite elements result, as well as motion amplitudes in comparison with a numerical boundary element model, provide valid validation of the present solution. With additional considerations of wave forces acting on the mooring lines, the drag dampening significantly decreases wave reflections and the motions of the structure at the high-frequency resonance. The magnitudes of the wave forces acting on the mooring lines can reach up to 12% of the incident wave forces. The study of the submerged depth of the structure indicates that the structure deployed nearer the free surface can induce bigger motions. The analytic solution is very much dependent on the geometry of the structure; however, the interaction formulation in this paper can be applied to practical problems for more complete considerations.

**Author Contributions:** Conceptualization, J.-F.L., C.-H.L.; methodology, J.-F.L., C.-T.C.; validation, C.-H.L.; writing—original draft preparation, J.-F.L., C.-H.L.; writing—review and editing, C.-T.C.; funding acquisition, C.-T.C., J.-F.L. All authors have read and agreed to the published version of the manuscript.

**Funding:** This research was funded by Ministry of Science and Technology, Taiwan, grant number MOST 104-2221-E-006-188.

**Acknowledgments:** Financial support partially by the Yancheng Institute of Technology under Grant Number XJ201750 and partially by the Ministry of Science and Technology, Taiwan, under Grant Number MOST 104-2221-E-006-188 are gratefully acknowledged.

**Conflicts of Interest:** The authors declare no conflict of interest.

## Appendix A. Definitions of Coefficients

Definitions of coefficients $k_n$, $k_{2n}$, $k_{4n}$, $\gamma_n$, $\mu_{n1}$, $\mu_{n2}$, $D_{2n}^2$, $F_{4n}^2$, $D_{2n}^3$, and $F_{4n}^3$.

$$\omega^2 = -gk_n \tan k_n h, \ n = 1, 2, \cdots \infty, \ k_0 = -iK \tag{A1}$$

$$\omega^2 = -gk_{2n} \tan k_{2n} d_1, n = 1, 2, \cdots \infty \tag{A2}$$

$$k_{4n} = n\pi/d_2, n = 1, 2, \cdots \infty \tag{A3}$$

$$\gamma_n = n\pi/2\ell, n = 1, 2, \cdots \infty \tag{A4}$$

$$\mu_{n1} = \gamma_n + \omega^2/g, n = 1, 2, \cdots \infty \tag{A5}$$

$$\mu_{n2} = \gamma_n - \omega^2/g, n = 1, 2, \cdots \infty \tag{A6}$$

$$D_{2n}^2 = \frac{i\omega s_2(1 - \cos 2\gamma_n \ell)}{\ell\gamma_n{}^2\left(\mu_{n1}e^{\gamma_n(-d_1-h)} - \mu_{n2}e^{-\gamma_n(-d_1+h)}\right)}, n = 1, 2, \cdots \infty \tag{A7}$$

$$F_{4n}^2 = \frac{i\omega s_2(1 - \cos 2\gamma_n \ell)}{\ell\gamma_n{}^2 \sinh\gamma_n d_2}, n = 1, 2, \cdots \infty \tag{A8}$$

$$D_{3n}^2 = \frac{-i\omega s_3(1 + \cos 2\gamma_n \ell)}{\gamma_n{}^2\left(\mu_{n1}e^{\gamma_n(-d_1-h)} - \mu_{n2}e^{-\gamma_n(-d_1+h)}\right)}, n = 1, 2, \cdots \infty \tag{A9}$$

$$F_{4n}^3 = \frac{-i\omega s_3(1 + \cos 2\gamma_n \ell)}{\gamma_n{}^2 \sinh \gamma_n d_2}, n = 1, 2, \cdots \infty \tag{A10}$$

## Appendix B. Wave Forces Acting on the Floating Structure

Wave forces acting on the floating structure in the direction of each degree of freedom can be calculated as:

$$\begin{aligned}
f_1 = -i\omega\rho \cdot e^{-i\omega t} \Big\{ &\int_{-h+d_2}^{-d_1} \Big[ \left(\phi^I + \phi_1^D\right)\big|_{x=-\ell} - \phi_3^D\big|_{x=\ell} \Big] dz \\
&+ s_1 \cdot \int_{-h+d_2}^{-d_1} \Big[ \phi_1^1\big|_{x=-\ell} - \phi_3^1\big|_{x=\ell} \Big] dz + s_3 \cdot \int_{-h+d_2}^{-d_1} \Big[ \phi_1^3\big|_{x=-\ell} - \phi_3^3\big|_{x=\ell} \Big] dz \Big\}
\end{aligned} \tag{A11}$$

$$\begin{aligned}
f_2 = -i\omega\rho \cdot e^{-i\omega t} \Big[ &\int_{-\ell}^{\ell} \left( \phi_4^D\big|_{z=-h+d_2} - \phi_2^D\big|_{x=-d_1} \right) dx \\
&+ s_2 \cdot \int_{-\ell}^{\ell} \left( \phi_4^2\big|_{z=-h+d_2} - \phi_2^2\big|_{z=-d_1} \right) dx \Big]
\end{aligned} \tag{A12}$$

$$\begin{aligned}
f_3 = \quad -i\omega\rho \cdot e^{-i\omega t} \Big\{ &\int_{-h+d_2}^{-d_1} (z - z_0) \Big[ \left(\phi^I + \phi_1^D\right)\big|_{x=-\ell} - \phi_3^D\big|_{x=\ell} \Big] dz \\
&- \int_{-\ell}^{\ell} x \Big[ \phi_4^D\big|_{z=-h+d_2} - \phi_2^D\big|_{z=-d_1} \Big] dx + s_1 \cdot \int_{-h+d_2}^{-d_1} (z - z_0) \Big[ \phi_1^1\big|_{x=-\ell} - \phi_3^1\big|_{x=\ell} \Big] dz \\
&- s_1 \cdot \int_{-\ell}^{\ell} x \Big[ \phi_4^1\big|_{z=-h+d_2} - \phi_2^1\big|_{z=-d_1} \Big] dx + s_3 \cdot \int_{-h+d_2}^{-d_1} (z - z_0) \Big[ \phi_1^3\big|_{x=-\ell} - \phi_3^3\big|_{x=\ell} \Big] dz \\
&- s_3 \cdot \int_{-\ell}^{\ell} x \Big[ \phi_4^3\big|_{z=-h+d_2} - \phi_2^3\big|_{z=-d_1} \Big] dx \Big\}
\end{aligned} \tag{A13}$$

Integrations shown in Equations (A11)–(A13) can be calculated, and the equations be rewritten, as:

$$\left\{ \begin{array}{c} f_1 \\ f_2 \\ f_3 \end{array} \right\} = \left[ f^R \right] \left\{ \begin{array}{c} s_1 \\ s_2 \\ s_3 \end{array} \right\} + \left\{ \begin{array}{c} f_1^D \\ f_3^D \\ f_3^D \end{array} \right\} \tag{A14}$$

where $\left[ f^R \right]$ and $\left\{ f^D \right\}$ can be expressed as:

$$\left[ f^R \right] = \begin{bmatrix} f_{11}^R & 0 & f_{13}^R \\ 0 & f_{22}^R & 0 \\ f_{31}^R & 0 & f_{33}^R \end{bmatrix} \tag{A15}$$

with the components expressed as:

$$f_{11}^R = \sum_{n=0}^{\infty} \left( B_{1n}^1 - A_{3n}^1 \right) \frac{\sin k_n(h - d_1) - \sin k_n d_2}{k_n} \tag{A16}$$

$$f_{13}^R = \sum_{n=0}^{\infty} \left( B_{1n}^3 - A_{3n}^3 \right) \frac{\sin k_n(h - d_1) - \sin k_n d_2}{k_n} \tag{A17}$$

$$\begin{aligned}
f_{22}^R = \quad &\left( 2\ell B_{40}^2 \right) + \sum_{n=1}^{\infty} \left( A_{4n}^2 + B_{4n}^2 \right) \cos k_{4n} d_2 \frac{\left(1 - e^{-2k_{4n}\ell}\right)}{k_{4n}} \\
&+ \sum_{n=1}^{\infty} F_{4n}^2 \cosh \kappa_{4n} d_2 \frac{(1 - \cos 2\kappa_{4n}\ell)}{\kappa_{4n}} - \sum_{n=0}^{\infty} \left( A_{2n}^2 + B_{2n}^2 \right) \frac{\left(1 - e^{-2k_{2n}\ell}\right)}{k_{2n}} \\
&- \sum_{n=1}^{\infty} D_{2n}^2 \left( \mu_{n1} e^{-\kappa_{2n}(d_1+h)} + \mu_{n2} e^{-\kappa_{2n}(-d_1+h)} \right) \frac{(1 - \cos 2\kappa_{2n}\ell)}{\kappa_{2n}}
\end{aligned} \tag{A18}$$

$$\begin{aligned}
f_{31}^R = \quad &\sum_{n=0}^{\infty} \left( B_{1n}^1 - A_{3n}^1 \right) C_n^{16} - \tfrac{2}{3} \ell^3 A_{40}^1 \\
&- \sum_{n=1}^{\infty} \left( A_{4n}^1 - B_{4n}^1 \right) \cos k_{4n} d_2 \cdot C_{4n}^{17} + \sum_{n=0}^{\infty} \left( A_{2n}^1 - B_{2n}^1 \right) C_{2n}^{17}
\end{aligned} \tag{A19}$$

$$
\begin{aligned}
f_{33}^R = & \sum_{n=0}^{\infty} \left(B_{1n}^3 - A_{3n}^3\right)C_n^{16} - \tfrac{2}{3}\ell^3 A_{40}^3 \\
& - \sum_{n=1}^{\infty} \left(A_{4n}^3 - B_{4n}^3\right)\cos k_{4n}d_2 \cdot C_{4n}^{17} - \sum_{n=1}^{\infty} F_{4n}^3 \cosh\kappa_n d_2 \cdot C_n^{18} \\
& + \sum_{n=0}^{\infty} \left(A_{2n}^3 - B_{2n}^3\right)C_{2n}^{17} - \sum_{n=1}^{\infty} D_{2n}^3\left(\mu_{n1}e^{-\kappa_{2n}(d_1+h)} + \mu_{n2}e^{-\kappa_{2n}(-d_1+h)}\right)C_n^{18}
\end{aligned}
\tag{A20}
$$

$$
\begin{aligned}
f_1^D = & \frac{igA^Ie^{-iK\ell}}{\omega\cosh Kh}\frac{\sinh K(h-d_1) - \sinh Kd_2}{K} \\
& + \sum_{n=0}^{\infty} \left(B_{1n}^D - A_{3n}^D\right)\cdot\frac{\sin k_n(h-d_1) - \sin k_n d_2}{k_n}
\end{aligned}
\tag{A21}
$$

$$
\begin{aligned}
f_2^D = & \ 2\ell B_{40}^D + \sum_{n=1}^{\infty}\left(A_{4n}^D + B_{4n}^D\right)\cos k_{4n}d_2\frac{\left(1 - e^{-2k_{4n}\ell}\right)}{k_{4n}} \\
& - \sum_{n=0}^{\infty}\left(A_{2n}^D + B_{2n}^D\right)\frac{\left(1 - e^{-2k_{2n}\ell}\right)}{k_{2n}}
\end{aligned}
\tag{A22}
$$

$$
\begin{aligned}
f_3^D = & \frac{igA^Ie^{-iK\ell}}{\omega\cosh Kh}C_0^{16} + \sum_{n=0}^{\infty}\left(B_{1n}^D - A_{3n}^D\right)C_n^{16} - \tfrac{2}{3}\ell^3 A_{30}^D \\
& + \sum_{n=1}^{\infty}\left(A_{4n}^D - B_{4n}^D\right)\cos k_{4n}d_2 \cdot C_{4n}^{17} + \left(A_{2n}^3 - B_{2n}^3\right)C_{2n}^{17}
\end{aligned}
\tag{A23}
$$

And the constants $C_n^{16}$, $C_{2n}^{17}$, $C_{4n}^{17}$, and $C_n^{18}$ are:

$$
\begin{aligned}
C_n^{16} = & \int_{-d_1}^0 (z - z_0)\cos[k_n(z+h)]dz \\
= & \frac{(h-d_2-z_0)\sin k_n d_2 - (d_1+z_0)\sin k_n(h-d_1)}{k_n} \\
& + \frac{\cos k_n(h-d_1) - \cos k_n d_2}{k_n^2}
\end{aligned}
\tag{A24}
$$

$$
\begin{aligned}
C_{2n,4n}^{17} = & \int_{-\ell}^{\ell} xe^{-k_{2n,4n}(x+\ell)}dx = -\int_{-\ell}^{\ell} xe^{k_{2n,4n}(x-\ell)}dx \\
= & \frac{1 - e^{-2k_{2n,4n}\ell}}{k_{2n,4n}^2} - \frac{\ell\left(1 + e^{-2k_{2n,4n}\ell}\right)}{k_{2n,4n}}
\end{aligned}
\tag{A25}
$$

$$
\begin{aligned}
C_n^{18} = & \int_{-\ell}^{\ell} x\sin\gamma_n(x+\ell)dx \\
= & \frac{\sin 2\gamma_n\ell}{\gamma_n^2} - \frac{\ell(1 + \cos 2\gamma_n\ell)}{\gamma_n^2}
\end{aligned}
\tag{A26}
$$

## Appendix C. Stiffness Matrix of the Mooring Springs

The components of the stiffness matrices for the springs $\overline{AB}$ and $\overline{CD}$ are calculated according to the geometrical orientations of the springs and can be expressed as:

$$
K_{11}^{\overline{AB}} = K_{11}^{\overline{CD}} = K_s\cos^2\theta
\tag{A27}
$$

$$
K_{12}^{\overline{AB}} = K_{21}^{\overline{AB}} = -K_{12}^{\overline{CD}} = -K_{21}^{\overline{CD}} = K_s\cos\theta\sin\theta
\tag{A28}
$$

$$
K_{13}^{\overline{AB}} = K_{31}^{\overline{AB}} = K_{13}^{\overline{CD}} = K_{31}^{\overline{CD}} = K_s\left[0.5(h-d_1-d_2)\cos^2\theta - \ell\cos\theta\sin\theta\right]
\tag{A29}
$$

$$
K_{22}^{\overline{AB}} = K_{22}^{\overline{CD}} = K_s\sin^2\theta
\tag{A30}
$$

$$
K_{33}^{\overline{AB}} = K_{33}^{\overline{CD}} = K_s[0.5(h-d_1-d_2)\cos\theta - \ell\sin\theta]^2
\tag{A31}
$$

## Appendix D. Wave Forces Acting on Mooring Lines

Wave forces acting on the mooring springs are calculated using the linearized Morison equation (Lee, 1994):

$$
dF^M = \frac{\rho C_{D\ell}D_S}{2}\left(U - \dot{\varsigma}\right)dS + \frac{\rho\pi C_M D_S^2}{4}\left(\dot{U} - \ddot{\varsigma}\right)dS
\tag{A32}
$$

where $\rho$ is fluid density, $D_S$ is the diameter of the spring, $U$ and $\dot{U}$ are the flow velocity and acceleration in the direction normal to the mooring line, $C_M$ is the added mass coefficient, $\dot{\varsigma}$ and $\ddot{\varsigma}$ are the velocity and acceleration of the mooring line, and the linear drag coefficient is expressed as:

$$C_{D\ell} = \frac{4C_D}{3\pi\omega} \frac{\int_{-h}^{-h+d_2} \left|U - \dot{\varsigma}\right|^3 dz}{\int_{-h}^{-h+d_2} \left|U - \dot{\varsigma}\right|^2 dz} \tag{A33}$$

Note that motions of the mooring lines are not known in priori until the problem solved: therefore, a complete solution will contain an iteration procedure until a 0.5% convergent criteria is reached. Furthermore, the springs are not subjected to forces in a transverse direction; therefore, the wave forces calculated are transferred to the attached points A and C. Thus,

$$F_A^M = \frac{\csc\theta}{d_2} \int_{-h}^{-h+d_2} \left[\frac{\rho C_{D\ell} D_S}{2}\left(U_1 - \dot{\varsigma}_{\overline{AB}}\right) + \frac{\rho\pi C_M D_S^2}{4}\left(\dot{U}_1 - \ddot{\varsigma}_{\overline{AB}}\right)\right] dz \tag{A34}$$

$$F_C^M = \frac{\csc\theta}{d_2} \int_{-h}^{-h+d_2} \left[\frac{\rho C_{D\ell} D_S}{2}\left(U_3 - \dot{\varsigma}_{\overline{CD}}\right) + \frac{\rho\pi C_M D_S^2}{4}\left(\dot{U}_3 - \ddot{\varsigma}_{\overline{CD}}\right)\right] dz \tag{A35}$$

where the subscripts 1 and 3 stand for regions 1 and 3, while subscripts $\overline{AB}$ and $\overline{CD}$ stand for the spring AB and CD. The corresponding expressions are:

$$\begin{aligned} U_1 = \ &-\left(\Phi_x^I + \Phi_{1x}^D + s_1 \cdot \Phi_{1x}^1 + s_2 \cdot \Phi_{1x}^2 + s_3 \cdot \Phi_{1x}^3\right)\sin\theta \\ &-\left(\Phi_z^I + \Phi_{1z}^D + s_1 \cdot \Phi_{1z}^1 + s_2 \cdot \Phi_{1z}^2 + s_3 \cdot \Phi_{1z}^3\right)\cos\theta \end{aligned} \tag{A36}$$

$$\begin{aligned} U_3 = \ &-\left(\Phi_{3x}^D + s_1 \cdot \Phi_{3x}^1 + s_2 \cdot \Phi_{3x}^2 + s_3 \cdot \Phi_{3x}^3\right)\sin\theta \\ &-\left(\Phi_{3z}^D + s_1 \cdot \Phi_{1z}^1 + s_2 \cdot \Phi_{3z}^2 + s_3 \cdot \Phi_{3z}^3\right)\cos\theta \end{aligned} \tag{A37}$$

With substitutions of Equations (A36) and (A37) into Equation (A33) and Equation (A35), one can obtain

$$F_A^M = F_A^{Mw} + F_A^{Ms} \tag{A38}$$

$$F_C^M = F_C^{Mw} + F_C^{Ms} \tag{A39}$$

in which

$$\begin{aligned} F_A^{Mw} = \ &\frac{\csc\theta}{d_2}\left(\frac{\rho C_{D\ell} D_S}{2} - \frac{i\omega\rho\pi D_S^2 C_M}{4}\right)\left\{\left(\frac{KgA^I e^{-iK\ell}\sin\theta}{\omega\cosh Kh}\right)\left(\frac{Kd_2\sinh Kd_2 - \cosh Kd_2 + 1}{K^2}\right)\right. \\ &+\left(\frac{iKgA^I e^{-iK\ell}\cos\theta}{\omega\cosh Kh}\right)\left(\frac{Kd_2\cosh Kd_2 - \sinh Kd_2}{K^2}\right) \\ &+\sin\theta\left[\sum_{n=0}^{\infty}\left(B_{1n}^D + s_1 B_{1n}^1 + s_2 B_{1n}^2 + s_3 B_{1n}^3\right)\left(\frac{-k_n d_2 \sin k_n d_2 - \cos k_n d_2 - 1}{k_n}\right)\right] \\ &+\cos\theta\left.\left[\sum_{n=0}^{\infty}\left(B_{1n}^D + s_1 B_{1n}^1 + s_2 B_{1n}^2 + s_3 B_{1n}^3\right)\left(\frac{-\sin k_n d_2 + k_n d_2 \cos k_n d_2}{k_n}\right)\right]\right\} \end{aligned} \tag{A40}$$

$$\begin{aligned} F_A^{Ms} = \ &\frac{\csc\theta}{d_2}\left(\frac{i\omega\rho C_{D\ell} D_S^2}{2} + \frac{\pi\rho C_M \omega^2 D_S^2}{4}\right) \\ &\left\{s_1\left(\frac{\sin\theta}{d_2}\right)\frac{d_2^3}{3} - s_2\frac{d_2^2}{2}\cos\theta - s_3\left[\frac{(h-d_1-d_2)\sin\theta}{2d_2}\frac{d_2^3}{3} + \ell\frac{d_2^2}{2}\cos\theta\right]\right\} \end{aligned} \tag{A41}$$

$$\begin{aligned} F_C^{Mw} = \ &\frac{\csc\theta}{d_2}\left(\frac{\rho C_{D\ell} D_S}{2} - \frac{i\omega\rho\pi D_S^2 C_M}{4}\right) \\ &\cdot\left\{\sin\theta\left[\sum_{n=0}^{\infty}\left(A_{3n}^D + s_1 A_{3n}^1 + s_2 A_{3n}^2 + s_3 A_{3n}^3\right)\left(\frac{k_n d_2 \sin k_n d_2 + \cos k_n d_2 - 1}{k_n}\right)\right]\right. \\ &+\cos\theta\left.\left[\sum_{n=0}^{\infty}\left(A_{3n}^D + s_1 A_{3n}^1 + s_2 A_{3n}^2 + s_3 A_{3n}^3\right)\left(\frac{\sin k_n d_2 - k_n d_2 \cos k_n d_2}{k_n}\right)\right]\right\} \end{aligned} \tag{A42}$$

$$F_C^{Ms} = \frac{\csc\theta}{d_2}\left(\frac{i\omega\rho C_D \ell D_S}{2} + \frac{\pi\rho C_M \omega^2 D_S^2}{4}\right)$$
$$\left\{ s_1\left(\frac{\sin\theta}{d_2}\right)\frac{d_2^3}{3} + s_2\frac{d_2^2}{2}\cos\theta - s_3\left[\frac{(h-d_1-d_2)\sin\theta}{2d_2}\frac{d_2^3}{3} + \ell\frac{d_2^2}{2}\cos\theta\right]\right\} \tag{A43}$$

Note that the spring $\overline{CD}$ is located at the lee side of the structure; therefore, there is no incident wave in the expression.

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
