# Peer review of "Mooring Drag Effects in Interaction Problems of Waves and Moored Underwater Floating Structures"

_jmse, doi:10.3390/jmse8030146_

Round 1
Reviewer 1 Report
This paper attempts to develop analytical solutions for mooring drag effects of under water structures. Introduction: The introduction part does not adequately describe the frame work, problems of the research and a background to the research. I think that the introduction part needs to restructured and some portions need to be rewritten in order to improve the following basic elements of it: general overview, applications (very important), problem definition, literature review, motivation, research gap, objective and overview of the manuscript clearly. Analytical solutions: The new value is very low and the application of this approach is rather limited. Addition of other possible nonlinear effects associated with the mooring dynamics to the problem would strengthen the present study. What about the effect of current and the combined wave-current interactions? The authors should discuss how this new analytical approach might help to resolve specific characteristics of mooring problems (i.e., improvements of model’s representation of the physical processes compared with previous numerical and theoretical models). Could the authors provide a brief discussion of the limitations of the present approach? Validation study: The details such as input parameters, boundary conditions and an object used are not presented clearly for the validation case. More clarification is needed for the validation, as this has to be determined case by case, depending on different boundary conditions and mooring types. In fact, even more clarification will be brought on the validation procedure. Results and discussion: The explanations and discussion on the new findings are insufficient. The results and discussion part is short and this part should provide more information regarding how your results fit into the existing research results and show the connections between your results and the literature reviewed earlier. A physical explanation, governing mechanisms and discussion of the model results is needed. Overall, the paper lacks critical and in-depth assessment/analysis of new results, which is crucial for achieving the purpose of the paper. How efficient the present mathematical approach works better than previous mathematical, theoretical and numerical models computationally in solving the physical problem. The conclusion part of numerical findings is too general.Author Response
Please see the attachment. Thanks.

Reviewer 2 Report
(1) The paper aims to present an analytic solution for 2D coupled mooring analysis of a rectangular structure. The analytic solution ,however, is limited to wave force computation. This should be made clear in the abstract and introduction.
(2) The mooring analysis seems too simple. It is claimed in the Conclusions (section 4) that the paper considers "complete interactions between waves and moored floating structures". This claim is not justified. A complete mooring analysis would consider mass distribution and geometry of the mooring lines including the computation of the hydrodynamic forces on the lines as they move. The coupled dynamics of mooring lines and vessel would also be considered in an analysis that claims to consider "complete interactions"
(3) In Appendix IV, line 306, the authors refer to the velocity and acceleration of the mooring line. Do the authors mean the components of these quantities normal to the line length at a particular point ?
(4) The title of the paper suggests that a coupled problem is considered. In the opinion of this reviewer, a "coupled problem" should include the coupling of vessel and mooring line dynamics.
Round 2
Reviewer 1 Report
A major part of the manuscript is dedicated to the problem formulation and of course, this provides a good overview of the formulation. However, the paper lacks critical and in-depth assessment of new model results. In addition, the authors did not clearly address my comments in the first round of review. The introduction part is better now, but it can still be improved by adding more practical applications.
The following questions still remain unanswered:
Analytical solutions: The new value is very low and the application of this approach is rather limited. Addition of other possible nonlinear effects associated with the mooring dynamics to the problem would strengthen the present study. What about the effect of current and the combined wave-current interactions? The authors should discuss how this new analytical approach might help to resolve specific characteristics of mooring problems (i.e., improvements of model’s representation of the physical processes compared with previous numerical and theoretical models). Could the authors provide a brief discussion of the limitations of the present approach? Validation study: The details such as input parameters, boundary conditions and a geometric shape of the floating object used are not presented clearly for the validation case. Are there any experimental data available to validate your analytical model results? It is always better to validate the numerical results with experimental data. More clarification is needed for the validation, in particular for mooring line forces, as this has to be determined case by case, depending on different boundary conditions and shapes of floating structures. Results and discussion: The explanations and discussion on the new findings are insufficient. The results and discussion part is short and this part should provide more information regarding how your results fit into the existing research results and show the connections between your results and the literature reviewed earlier. A physical explanation, governing mechanisms and discussion of the model results is needed. Overall, the paper lacks critical and in-depth assessment/analysis of new model results, which is crucial for achieving the purpose of the paper. How efficient the present mathematical approach works better than previous mathematical, theoretical and numerical models computationally in solving the physical problem. The conclusion part can still be improved by adding meaningful findings, limitations and future work. The manuscript requires a thorough proof reading in order to improve the readability of the manuscript.
Reviewer 2 Report
English language corrections required
Author Response
Point 1: English language corrections required.
Response 1: We have talked to assistant Editor that we agreed to have English Editing.
Round 3
Reviewer 1 Report
The authors have satisfactorily addressed all my comments in the revised manuscript. However, the manuscript requires a thorough proof reading in order to improve the readability of the manuscript.
I have no further comments on the manuscript.
Author Response
Comments and suggestions: The manuscript requires a through proof reading in order to improve the readability.
Response: We have talked to the assistant editor, and we agreed to have English Editing to improve the manuscript.